



# Modelling the growth of atmospheric nitrous oxide using a global hierarchical inversion

Angharad C. Stell[1], Michael Bertolacci[2], Andrew Zammit-Mangion[2], Matthew Rigby[3], Paul J. Fraser[4], Christina M. Harth[5], Paul B. Krummel[4], Xin Lan[6,7], Manfredi Manizza[5], Jens Mühle[5], Simon O'Doherty[3], Ronald G. Prinn[8], Ray F. Weiss[5], Dickon Young[7], and Anita L. Ganesan[1]

[1]School of Geographical Sciences, University of Bristol, Bristol, UK
[2]School of Mathematics and Applied Statistics, University of Wollongong, Wollongong, Australia
[3]School of Chemistry, University of Bristol, Bristol, UK
[4]Climate Science Centre, CSIRO Oceans and Atmosphere, Aspendale, Australia
[5]Scripps Institution of Oceanography, University of California, San Diego, USA
[6]National Oceanic and Atmospheric Administration, Earth System Research Laboratory, Boulder, USA
[7]University of Colorado, Cooperative Institute for Research in Environmental Sciences, Boulder, USA
[8]Center for Global Change Science, Massachusetts Institute of Technology, Cambridge, USA

**Correspondence:** Angharad C. Stell (a.stell@bristol.ac.uk), Anita L. Ganesan (anita.ganesan@bristol.ac.uk)

**Abstract.**

Nitrous oxide is a potent greenhouse gas and ozone depleting substance, whose atmospheric abundance has risen throughout the contemporary record. In this work, we carry out the first global hierarchical Bayesian inversion to solve for nitrous oxide emissions, which includes prior emissions with non-Gaussian distributions and model errors, in order to examine the drivers of the atmospheric surface growth rate. We show that both meteorology and emissions are key drivers of variations in the surface nitrous oxide growth rate between 2011 and 2020. We derive increasing global nitrous oxide emissions, which are mainly driven by emissions between $0°$ and $30°$ N, with the highest emissions recorded in 2020. Our mean global total emissions for 2011–2020 of 17.2 (16.7–17.7 at the 95% credible intervals) TgN yr$^{-1}$, comprising of 12.0 (11.2–12.8) TgN yr$^{-1}$ from land and 5.2 (4.5–5.9) TgN yr$^{-1}$ from ocean, agrees well with previous studies, but we find that emissions are poorly constrained for some regions of the world, particularly for the oceans. The prior emissions used in this and other previous work exhibit a seasonal cycle in the Northern Hemisphere extra-tropics that is out of phase with the posterior solution, and there is a substantial zonal redistribution of emissions from the prior to the posterior. Correctly characterising the uncertainties in the system, for example in the prior emission fields, is crucial to be able to derive posterior fluxes that are consistent with observations. In this hierarchical inversion, the model-measurement discrepancy and the prior flux uncertainty are informed by the data, rather than solely through expert judgment. We show cases where this framework provides different plausible adjustments to the prior fluxes compared to inversions using widely adopted, fixed uncertainty constraints.





## 1 Introduction

Nitrous oxide (N$_2$O) is an important greenhouse gas (GHG) that contributes substantially to the increase in radiative forcing of climate by anthropogenic activities (Myhre et al., 2013; Etminan et al., 2016). Additionally, considering ozone depletion potential-weighted anthropogenic emissions, nitrous oxide is currently the single greatest ozone depleting substance (Ravishankara et al., 2009). The amount of nitrous oxide in the atmosphere has risen from about 290 ppb in 1940 to 333 ppb in 2020 (Park et al., 2012; Prinn et al., 2000, 2018; Dlugokencky et al., 2021). This rise is predominantly due to increasing agricultural emissions (Davidson, 2009; Syakila and Kroeze, 2011; Tian et al., 2019). The natural sources of nitrous oxide are natural soils, biomass burning, and oceans, which are all highly uncertain in magnitude and distribution (e.g., Solazzo et al., 2021; Ciais et al., 2013). Nitrous oxide is only slowly removed from the atmosphere by photolysis and reaction with excited oxygen atoms (O($^1$D)) in the stratosphere, resulting in a lifetime of about 120 years (Ko et al., 2013; Prather et al., 2015).

The atmospheric abundance of nitrous oxide is monitored by the National Oceanic and Atmospheric Administration (NOAA) (Dlugokencky et al., 2021; Sweeney et al., 2021) and the Advanced Global Atmospheric Gases Experiment (AGAGE) (Prinn et al., 2000, 2018). Fig. 1 shows the atmospheric surface nitrous oxide growth rate from 2011 to 2020 based on these observations. From mid–2017 until 2019 the abundance of nitrous oxide was growing fastest in the Southern Hemisphere. Since 2000, this is only the second extended time period where the surface growth rate was led by the Southern Hemisphere. This pattern may be explained by increasing emissions within this region (Thompson et al., 2019; Tian et al., 2020; Patra et al., 2022), or by meteorology, which has been shown to be a key driver of the growth rate of surface nitrous oxide mole fraction (Ray et al., 2020; Ruiz et al., 2021).





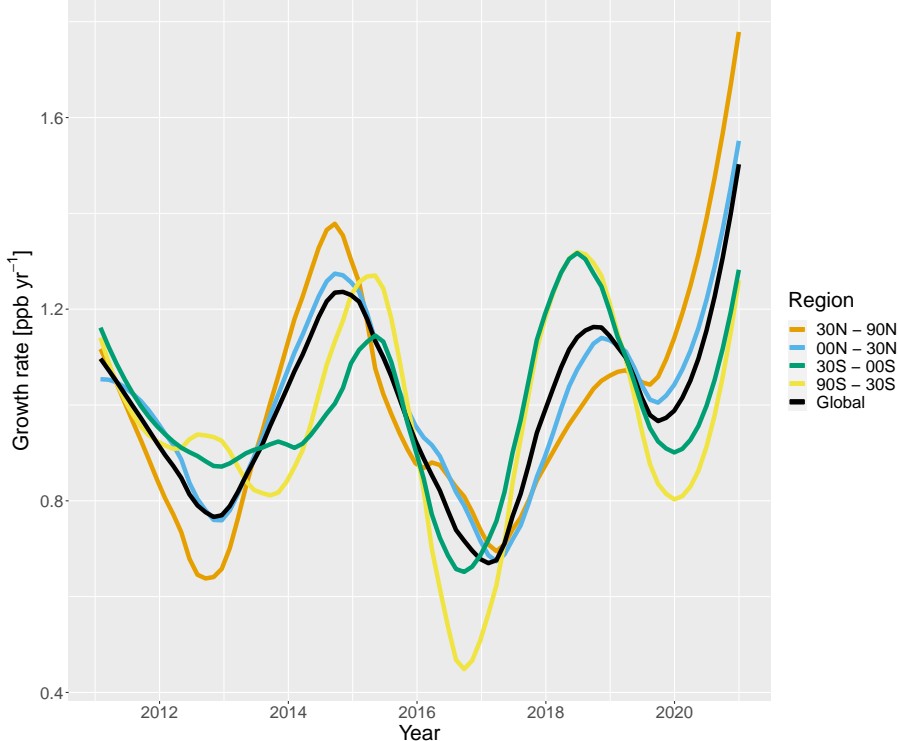

**Figure 1.** The observed atmospheric surface nitrous oxide growth rate derived from the AGAGE and NOAA networks for each month of 2011 to 2020 globally (black line) and in four latitude bands (coloured lines). The observations included are detailed in Sect. 2.1, and are combined into latitude band and global totals by weighting by the cosine of the latitude. The surface growth rate is calculated as the difference in the mole fraction between the month in the displayed year and the year before, and has been smoothed using a LOESS (locally weighted smoothing) algorithm with a span of 0.3.

Previous global nitrous oxide inversions that estimated emissions from atmospheric mole fraction data include Wells et al. (2015, 2018), Thompson et al. (2019), Tian et al. (2020), and Patra et al. (2022). The latter three investigated decadal scale emissions trends, finding that global nitrous oxide emissions have risen over the last two decades, with Thompson et al. (2019) attributing this rise to agricultural soils as a result of a non-linear relationship between N-input and nitrous oxide emissions.

    The agreement between previous inversion studies demonstrates that global total nitrous oxide emissions are well con-
strained by observations at around 17 TgN yr$^{-1}$. However, there is considerable variation on the regional scale. For example, one inversion setup in Thompson et al. (2019) derives oceanic emissions of 7.2 TgN yr$^{-1}$ over 1998–2016, whereas Patra et al. (2022) derives oceanic emissions of 2.8 TgN yr$^{-1}$ over 2000–2019. The discrepancy is also seen in developed land regions. For example, Wells et al. (2018) derives very different emissions for Europe (0.43–1.05 TgN yr$^{-1}$) depending on the inversion setup. These discrepancies suggest that more atmospheric observations are required to constrain fluxes at the regional scale in
global inversions.





The limited ability of atmospheric observations within global inversions to partition emissions at the regional-scale means that the "bottom-up" inventory and process modelling estimates used as prior estimates for the "top-down" inversion methods could strongly influence the inversion results. The majority of nitrous oxide emissions come from poorly understood microbial processes in the soil (Butterbach-Bahl et al., 2013), which are controlled by temperature, moisture, nitrogen inputs, and other

environmental factors. Most of the remaining emissions are oceanic, and are also derived from microbial processes. Marine nitrous oxide emissions additionally require knowledge of air-sea exchange (Nevison et al., 1995; Manizza et al., 2012; Yang et al., 2020). The complex and poorly understood nature of nitrous oxide emissions means that uncertainties in the prior estimates are difficult to characterise. For example, the posterior solutions of several previous inversions have substantially altered seasonal cycles compared to bottom-up studies (e.g., Thompson et al., 2014; Nevison et al., 2018; Wells et al., 2018).

This discrepancy is thought to be due to missing freeze-thaw processes or fertiliser application timings in process models (Wagner-Riddle et al., 2017; Nevison et al., 2018; Wells et al., 2018), or inaccuracies in top-down estimates due to model transport (Nevison et al., 2007; Thompson et al., 2014).

Here, for the first time, we use a hierarchical Bayesian global inversion framework to estimate nitrous oxide emissions from 2011–2020. Previous studies investigating nitrous oxide have used either an analytical Bayesian inversion framework (e.g.,

Thompson et al., 2019; Tian et al., 2020; Patra et al., 2022) or a four-dimensional variational (4D-Var) method (e.g., Wells et al., 2015, 2018; Thompson et al., 2019; Tian et al., 2020). Our hierarchical Bayesian inversion framework is advantageous as both analytical and 4D-Var atmospheric inversions require specification of uncertainties on the prior fluxes and model error, both often assumed to be Gaussian, which are determined by "expert judgment". Incorrectly specified uncertainties can significantly impact the posterior solution (Ganesan et al., 2014). The hierarchical inversion addresses this by using hyper-parameters to

explore a range of possible prior uncertainties. Additionally, using Markov chain Monte Carlo (MCMC) allows the use of non-Gaussian flux distributions, which cannot easily be implemented in analytical inversion systems. These distributions are useful for gases such as nitrous oxide, as we expect land emissions to be only positive.

## 2  Methods

### 2.1  Atmospheric observations

The atmospheric observations used in this work are surface measurements from 45 stations, which are listed in Table 1 and mapped in Fig. 2. These observations were made by the Advanced Global Atmospheric Gases Experiment (AGAGE; Prinn et al., 2000, 2018) and as part of two National Oceanic and Atmospheric Administration (NOAA) programs: the Halocarbons and other Atmospheric Trace Species (HATS) and the Carbon Cycle Greenhouse Gases (CCGG; Dlugokencky et al., 2021). The NOAA stations were selected based on two criteria: (i) they have nitrous oxide records for at least six of the years in the

target time period (2010–2020) to prevent temporal inconsistencies in the inferred fluxes as stations come in and out of service, (ii) they are not heavily influenced by local nitrous oxide sources, which is determined by visual comparison of the GEOS-Chem base run (Sect. 2.2) and the observations. This filtering is necessary because the model resolution is too coarse to simulate



local effects. High-frequency AGAGE data are similarly filtered to only include samples representative of background air. The background air samples are identified using the Lagrangian model, NAME (Numerical Atmospheric dispersion Modelling

Environment; Jones et al., 2007), as samples where the proportion of air from the surrounding grid cells, populated areas, and the upper troposphere is low (Arnold et al., 2018).

The NOAA and AGAGE networks are on different calibration scales. To prevent this from affecting the inversion, we harmonise the networks by rescaling the AGAGE data using the method of Wells et al. (2018). This is done by using measurements from locations where both AGAGE and NOAA data are available (Cape Grim, Mace Head, Ragged Point, Tutuila, and Trinidad

Head). AGAGE and NOAA measurements made within 15 minutes of each other are matched, and the average ratio between the matched AGAGE and NOAA measurements (AGAGE measurements / NOAA measurements) was found to be 1.0015. This ratio is used to rescale the AGAGE data. A single ratio was used for the whole time period as there was no evidence of a trend in this value over time.

Following the calibration adjustment, for each station we compute monthly averages of the raw measurements, along with

the standard deviation of the measurements in that month. The monthly averages are used as observations for the inversion, and the standard deviations are used as the measurement error component of the total error budget for each observation. This is a conservative estimate of the measurement error that allows for the possibility of very high correlation between measurements within each month. If there is only one sample at a station for a month, or the calculated standard deviation is smaller than the median instrumental uncertainty reported by the data provider over all stations that month, then the median reported

instrumental uncertainty is used. This equates to median measurement uncertainties of 0.26 ppb. The other component of the total error budget for each observation is model error, which we discuss in Sect. 2.3.3.





**Table 1.** Surface stations included in the inversion. Locations of the stations are shown in Fig. 2.

| Station | Network | Location |
| --- | --- | --- |
| ALT | NOAA | Nunavut, Cananda |
| ASC | NOAA | Ascension Island, United Kingdom |
| ASK | NOAA | Assekrem, Algeria |
| AZR | NOAA | Terceira Island, Azores, Portugal |
| BAO | NOAA | Boulder Atmospheric Observatory, Colorado, United States |
| BHD | NOAA | Baring Head Station, New Zealand |
| BMW | NOAA | Tudor Hill, Bermuda, United Kingdom |
| BRW | NOAA | Barrow Atmospheric Baseline Observatory, Alaska, United States |
| CBA | NOAA | Cold Bay, Alaska, United States |
| CGO | NOAA and AGAGE | Cape Grim, Tasmania, Australia |
| CHR | NOAA | Christmas Island, Republic of Kiribati |
| CPT | NOAA | Cape Point, South Africa |
| CRV | NOAA | Carbon in Arctic Reservoirs Vulnerability Experiment (CARVE), Alaska, United States |
| CRZ | NOAA | Crozet Island, France |
| DRP | NOAA | Drake Passage |
| EIC | NOAA | Easter Island, Chile |
| GMI | NOAA | Mariana Islands, Guam |
| HBA | NOAA | Halley Station, Antarctica, United Kingdom |
| ICE | NOAA | Storhofdi, Vestmannaeyjar, Iceland |
| IZO | NOAA | Izana, Tenerife, Canary Islands, Spain |
| KEY | NOAA | Key Biscayne, Florida, United States |
| KUM | NOAA | Cape Kumukahi, Hawaii, United States |
| LLN | NOAA | Lulin, Taiwan |
| LMP | NOAA | Lampedusa, Italy |
| MBO | NOAA | Mt. Bachelor Observatory, Oregon, United States |
| MEX | NOAA | High Altitude Global Climate Observation Center, Mexico |
| MHD | NOAA and AGAGE | Mace Head, County Galway, Ireland |
| MID | NOAA | Sand Island, Midway, United States |
| MLO | NOAA | Mauna Loa, Hawaii, United States |
| MWO | NOAA | Mt. Wilson Observatory, California, United States |
| NAT | NOAA | Farol De Mae Luiza Lighthouse, Brazil |
| NMB | NOAA | Gobabeb, Namibia |
| NWR | NOAA | Niwot Ridge, Colorado, United States |
| PSA | NOAA | Palmer Station, Antarctica, United States |
| RPB | NOAA and AGAGE | Ragged Point, Barbados |
| SEY | NOAA | Mache Island, Seychelles |
| SMO | NOAA and AGAGE | Tutuila, American Samoa |
| SPO | NOAA | South Pole, Antarctica, United States |
| SUM | NOAA | Summit, Greenland |
| SYO | NOAA | Syowa Station, Antarctica, Japan |
| THD | NOAA and AGAGE | Trinidad Head, California, United States |
| TIK | NOAA | Hydrometeorological Observatory of Tiksi, Russia |
| USH | NOAA | Ushuaia, Argentina |
| UUM | NOAA | Ulaan Uul, Mongolia |
| ZEP | NOAA | Ny-Alesund, Svalbard, Norway and Sweden |





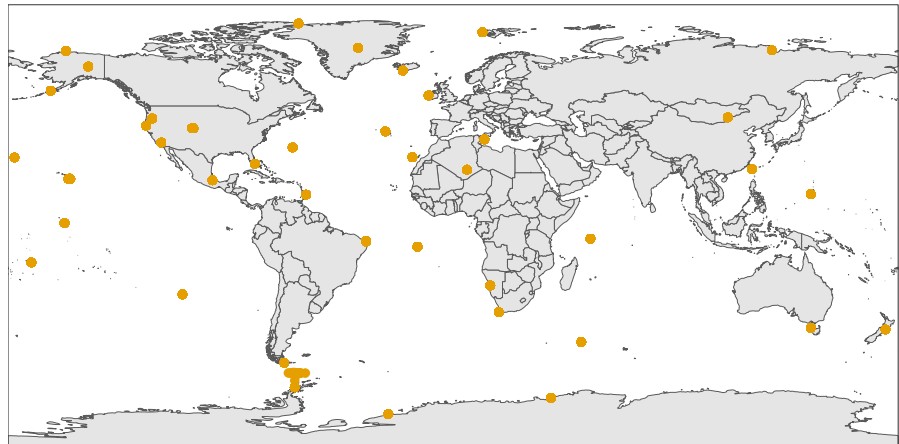

**Figure 2.** The locations of the observations used in this work.

## 2.2 Nitrous oxide prior emissions and model simulations

We run model simulations to link the observations of nitrous oxide mole fractions to emissions. The simulations use the GEOS-Chem chemical transport model (http://acmg.seas.harvard.edu/geos/), version 13.0.0, run with a horizontal resolution
of $4° \times 5°$ and 72 vertical levels from the surface to 0.01 hPa. The time steps are 10 minutes for transport and 20 minutes for chemistry and emissions, and use MERRA-2 meteorology (Gelaro et al., 2017).

The prior emissions we use are a combination of anthropogenic emissions from EDGARv5.0 (Crippa et al., 2019), natural soil emissions from Saikawa et al. (2013), oceanic emissions using output from the ocean model ECCO2-Darwin (Ganesan et al., 2020), and biomass burning emissions from GFED4 (Randerson et al., 2017). The sources and their temporal resolution
are given in Table 2. The emissions from soil, oceans, and biomass burning are turned into monthly climatologies by taking the average in each calendar month in the dataset. The stratospheric loss of nitrous oxide by photolysis and reaction with O($^1$D) is taken from archived monthly loss frequencies from the Global Modelling Initiative (GMI; Rotman et al., 2001).

To derive an initial condition for the nitrous oxide mole fraction, we run a spin-up simulation for 10 years using repeating 2009 emissions and meteorology, starting from an atmosphere with a constant nitrous oxide mole fraction. The resulting initial
condition field matches surface nitrous oxide observations to within a few ppb and also gives a nitrous oxide lifetime of 120 years, in good agreement with Ko et al. (2013) and Prather et al. (2015). A "base" simulation is then run for 2010–2020 with time-varying meteorology and prior emissions. Further simulations using fluxes that are perturbed from the prior are used to construct basis functions, which are described in Sect. 2.3.1. In order to compare the model simulations to the observations, the modelled mole fraction is sampled at the latitude, longitude, altitude, and time of the measurements described in Sect. 2.1.
Monthly mean values are created from these samples using the same method as for the observations, as in Sect. 2.1.





**Table 2.** Emissions inventories used for prior emissions in this work.

| Source | Reference | Temporal resolution | Years |
|--------|-----------|---------------------|-------|
| Anthropogenic | Crippa et al. (2019) | annual | 2009-2015 (2016-2020 is 2015 repeating) |
| Natural soils | Saikawa et al. (2013) | monthly | climatology |
| Oceans | Ganesan et al. (2020) | monthly | climatology |
| Biomass burning | Randerson et al. (2017) | monthly | climatology |

## 2.3 WOMBAT inversion framework

The inversion uses a hierarchical Bayesian inversion framework called WOMBAT (the WOllongong Methodology for Bayesian Assimilation of Trace-gases), which has previously been used for estimating carbon dioxide emissions from satellite data (Zammit-Mangion et al., 2022). The WOMBAT framework was developed to reduce the problem of model misspecifica-
tion caused by issues such as: an inaccurate prior flux field and uncertainty; retrieval biases for satellite data; and possible spatio-temporal correlations in the measurement error (Zammit-Mangion et al., 2022). WOMBAT tackles these problems by: specifying prior distributions on the uncertainty in the prior fluxes; modelling biases in the mole fraction data; adding a spatio-temporally correlated component of variability to the measurement error; and propagating uncertainty on all unknowns within a fully Bayesian statistical framework where inference is made using MCMC. This framework therefore provides a more statis-
tically rigorous approach than many previous atmospheric flux inversions. A complete description of the framework for carbon dioxide inversions is given by Zammit-Mangion et al. (2022). Here, we provide a brief description of the modified framework used here.

This work is set up as for carbon dioxide in Zammit-Mangion et al. (2022), with four exceptions:

1. No bias in observations and no correlation in model-measurement discrepancy are considered, as the monthly data used
are less likely to have correlated errors than the higher frequency data used in Zammit-Mangion et al. (2022).

2. The fluxes are described by a Gaussian prior distribution truncated at zero to prevent negative emissions from land.

3. Fluxes are estimated using a 3-year moving window to reduce the computational cost of the inversion.

4. The autocorrelation between flux scaling factors is assumed to be zero due to the timing of the prior seasonal cycle, discussed in Sect. 3.3.2.

### 2.3.1 The flux process model

The true flux of nitrous oxide ($Y_1$) is modelled as the prior flux ($Y_1^0$) plus a sum of $r$ flux basis functions ($\phi_j$), which are weighted by scaling factors ($\alpha$). In this study, there are 3 036 flux basis functions spanning the 23 TransCom regions (Fig. S4;





see Gurney et al., 2002) and the 132 months of the study period. The scaling factors ($\alpha$) are estimated in the inversion. The flux process model may be written as

$$Y_1(\boldsymbol{s}, t) = Y_1^0(\boldsymbol{s}, t) + \sum_{j=1}^{r} \phi_j(\boldsymbol{s}, t)\alpha_j + v_1(\boldsymbol{s}, t), \tag{1}$$

where $\boldsymbol{s}$ is the spatial location, $t$ denotes time, and $v_1$ is an error term. The basis functions are set equal to the prior emissions in their corresponding region and month, and zero elsewhere. Consequently, excluding the error term, the true flux in a region is modelled as a scaling of the prior flux in that region. The error term $v_1$ accommodates deviations between the true flux spatio-temporal patterns and those in the prior emissions.

### 2.3.2 The mole fraction process model

Like the flux field, the mole fraction field has a basis function representation, where each flux basis function has a corresponding response function representing the impact of the prior emissions in a TransCom region and a month on the atmospheric mole fraction field. The true mole fraction ($Y_2$) at space-height-time location ($\boldsymbol{s}, h, t$) is modelled as the prior expectation of the mole fraction field derived by the chemical transport model ($Y_2^0$) plus a sum of the $r$ response functions ($\psi_j$) which are weighted by 150 the same scaling factors ($\alpha$) that appear in Eq. 1. The resulting mole fraction process model is

$$Y_2(\boldsymbol{s}, h, t) = Y_2^0(\boldsymbol{s}, h, t) + \sum_{j=1}^{r} \psi_j(\boldsymbol{s}, h, t)\alpha_j + v_2(\boldsymbol{s}, h, t), \tag{2}$$

where $v_2$ amalgamates spatio-temporal errors from the use of low-dimensional basis functions and of a chemical transport model that does not simulate transport and chemistry perfectly. To construct each response function, we run a perturbed model simulation where the prior fluxes are doubled in that region and month, then subtract the simulated base mole fraction field 155 from the field simulated under the perturbation. The perturbed simulations are run for two years, past which the response function is assumed to be constant in each grid cell.

### 2.3.3 The mole fraction data model

The data used to constrain the nitrous oxide fluxes are the monthly mean mole fractions described in Sect. 2.1. The $i^{\text{th}}$ measured value ($Z_{2,i}$) at space-height-time location ($\boldsymbol{s}_i, h_i, t_i$) differs from the actual true mole fraction of the atmosphere ($Y_2$) by the 160 measurement error ($\epsilon_i$):

$$Z_{2,i}(\boldsymbol{s}, h, t) = Y_2(\boldsymbol{s}_i, h_i, t_i) + \epsilon_i. \tag{3}$$

Substituting Eq. 2 into Eq. 3 yields the relationship between the scaling factors ($\alpha$) and the measurements. The measurements are grouped by observation station into groups $g = 1, \ldots, n_g$ with similar error properties, and collected into vectors $\mathbf{Z}_{2,g}$. There





are 45 observation stations in this work, and so $n_g$ is 45 in this case. The model for the $g^{\text{th}}$ group can be written in matrix-vector

form as

$$\mathbf{Z}_{2,g} = \mathbf{Y}^0_{2,g} + \mathbf{\Psi}_g \boldsymbol{\alpha} + \boldsymbol{\xi}_g, \tag{4}$$

where the error term $v_2$ and the measurement error $\epsilon_i$ have been amalgamated into an overall model-measurement discrepancy

term, $\boldsymbol{\xi}_g$.

We assume that the elements of $\boldsymbol{\xi}_g$ are distributed as independent Gaussians with mean zero and variance as follows. Let $\xi_i$

be the model-measurement discrepancy term for observation $i$ in group $g$. We set the variance of $\xi_i$, the square root of which

we call the error budget for the observation, to

$$\text{var}(\xi_i) = \gamma_g^{-1}(\sigma_i^2 + \tau_i^2), \tag{5}$$

where $\gamma_g > 0$, $\sigma_i$ is the measurement error (Sect. 2.1), and $\tau_i$ is the model error. The term $\gamma_g$ is a station-specific (or equivalently,

group $g$ specific) error budget scaling factor which is estimated in the inversion. We assign the model error $\tau_i$ as follows. First,

we calculate the standard deviation in the mole fraction of a simulation run with the prior nitrous oxide emissions in the nine

horizontal model grid cells surrounding each observation. We then set $\tau_i$ to be the median standard deviation of the nine grid

cells for each month at each station. The median (over all sites and all months) measurement uncertainty, model uncertainty,

and overall error budget are 0.26 ppb, 0.08 ppb, and 0.27 ppb, respectively. This overall error budget is at the lower end of the

values seen in other recent nitrous oxide inversions (Thompson et al., 2019; Tian et al., 2020; Patra et al., 2022).

The estimated model errors are likely too small, as their construction considers only spatial variability (Chen and Prinn,

2006), and ignores other errors, such as those in atmospheric transport. One benefit of our approach, is that the scaling factor

$\gamma_g$ adjusts the error budgets for the station until they better match the scale of the errors seen in the inversion. We reflect this in

the prior distribution for $\gamma_g$, which is described in the next section.

### 2.3.4 The parameter model

In Zammit-Mangion et al. (2022) the scaling factors ($\boldsymbol{\alpha}$) are assigned a multivariate Gaussian prior. In this work, to constrain

the emissions to be non-negative, we instead use as prior the truncated Gaussian distribution $\boldsymbol{\alpha} \sim \text{TruncGau}(\mathbf{0}, \mathbf{\Sigma}_\alpha, F_\alpha)$,

where $\text{TruncGau}(\boldsymbol{\mu}, \mathbf{\Sigma}, F)$ denotes a multivariate Gaussian distribution with mean $\boldsymbol{\mu}$ and covariance $\mathbf{\Sigma}$, and values of $\boldsymbol{\alpha}$ are

constrained to the region $F$. The precision matrix $\mathbf{Q}_\alpha \equiv \mathbf{\Sigma}_\alpha^{-1}$ is diagonal with the diagonal equal to $\boldsymbol{w}$, and the truncation

region $F_\alpha$ is set such that $\alpha \geq -1$. This constrains the posterior emissions to have sign equal to that of the prior emissions at

every point in space and time. The elements of $\boldsymbol{w}$ are assigned independent gamma distributions with shape parameter of 4 and

rate parameter of 0.7 (i.e. $\text{Ga}(4, 0.7)$). The prior mean of an element of $\boldsymbol{w}$ is thus 5.7, corresponding to a standard deviation of

0.4 (i.e., a $1\sigma$ uncertainty of 40 % uncertainty) on the scaling factors.





The error budget scaling factors ($\gamma_g$) are given independent prior distributions of $\mathrm{Ga}(2.4, 5.4)$. This distribution has 5 % and 95 % percentiles of 0.1 and 1.0, respectively. This corresponds to the square of the error budget being between 1 and 10 times its nominal value of $(\sigma_i^2 + \tau_i^2)$, which reflects the belief that the model error is likely to be underestimated.

The relationship between the variables is summarised in Fig. 3 and the unknown parameters and their prior distributions are summarised in Table 3.

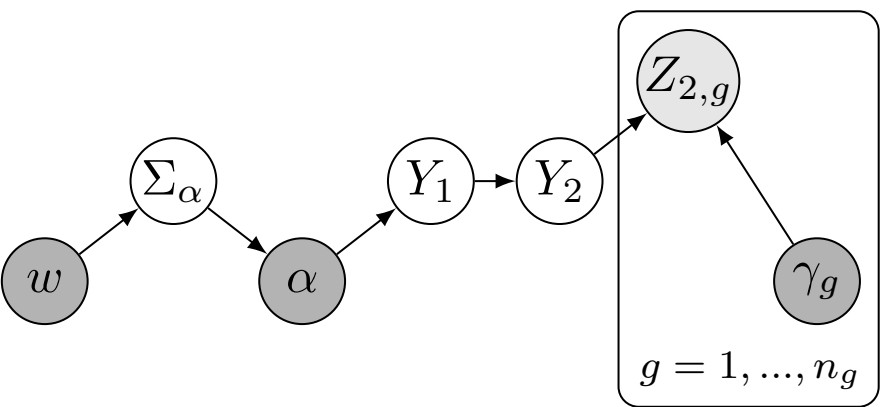

**Figure 3.** Graphical model summarising the relationship between the unknown parameters that are optimised in the inversion (dark grey, bottom row), intermediate variables (white, middle row), and the atmospheric mole fraction grouped by station (light grey, top row).

**Table 3.** Parameters optimised in the inversion, along with their prior distributions. The values used in the measurement-error variance inflation factors' prior are given to one decimal place.

| Parameter | Prior |
|---|---|
| Flux scaling factor's precisions ($w$) | Gamma(4, 0.7) |
| Flux scaling factors ($\boldsymbol{\alpha}$) | Gaussian($\mathbf{0}, \boldsymbol{\Sigma}_\alpha$), truncated so $\alpha \geq -1$ |
| Error budget scaling factors ($\gamma_g$) | Gamma(2.4, 5.4) |

### 2.3.5 Estimation of unknown parameters

The joint posterior distribution over the unknown parameters $\boldsymbol{\alpha}$, $\boldsymbol{w}$, and $\boldsymbol{\gamma}$ is sampled using the Gibbs sampler described by Zammit-Mangion et al. (2022), with the step to sample $\boldsymbol{\alpha}$ modified to use the method of Pakman and Paninski (2014) to accommodate the truncated Gaussian prior. MCMC, of which a Gibbs sampler is one form, generates samples from a target distribution by simulating a Markov chain that has the target distribution as its equilibrium distribution. MCMC is beneficial as it can be used to characterise distributions that are non-Gaussian.





The method of Zammit-Mangion et al. (2022) is too computationally expensive to run over the long time period in this study.
We instead use a moving window approach. This involves estimating the unknown parameters over a three year window (say, 2010–2012) and keeping only the parameter values inferred for the middle year (2011). This allows the first year to account for spin-up effects and the last year to contribute observations to the middle year. The next moving window (2011–2013) is then run with the prior mole fraction field for the start of the first year set to the posterior estimate from the previous window. For the last year, 2020, we use the estimates from a shorter 2019–2020 window, so fluxes from this year are more uncertain.

The results from the moving window approach were compared with varying length windows. Since the sensitivity of observations to a perturbation in the fluxes is nearly constant a year after the perturbation ceases, there was little difference between the flux scaling factors inferred by the three year moving window inversion and longer length windows. Therefore, we decided to use a three year window to minimise computational expense while maintaining accuracy.

## 2.4 Analytical inversion framework

To assess the departures from previous work caused by using a hierarchical inversion, an analytical inversion is also run for comparison. The observations, error budgets, prior fluxes, and prior flux uncertainty are set up as for the WOMBAT inversion framework (Sect. 2.3), with the exception that the prior flux distribution is Gaussian rather than a truncated Gaussian. There is no adjustment of the prior flux uncertainty or the error budgets in the inversion, and since the analytical inversion is far less computationally expensive, the inversion can be run for the whole time period without the moving window approach (Sect.
220 2.3.5).

In the analytical inversion, the optimal flux scaling factors are found using the linear least squares approach described by Tarantola (2005), which is outlined briefly here. The centre of the posterior Gaussian ($\boldsymbol{\alpha}$) is given by

$$\boldsymbol{\alpha} = \boldsymbol{\Sigma}_{\alpha} \boldsymbol{H}^t (\boldsymbol{H} \boldsymbol{\Sigma}_{\alpha} \boldsymbol{H}^t + \boldsymbol{\Sigma}_{\xi})^{-1} (\boldsymbol{Z}_2 - \boldsymbol{Y}_2^0), \tag{6}$$

where $\boldsymbol{\Sigma}_{\alpha}$ is the covariance matrix of $\boldsymbol{\alpha}$, $\boldsymbol{H}$ is the transport matrix (which transforms fluxes into modelled mole fractions),
$\boldsymbol{\Sigma}_{\xi}$ is the covariance matrix for the observations, $\boldsymbol{Z}_2$ is the observations, and $\boldsymbol{Y}_2^0$ is the modelled mole fraction using the prior emissions. The hyper-parameters $w$ and $\gamma$ are not solved for in this inversion, they are instead fixed values of 5.7 and 1, respectively. This results in $\boldsymbol{\Sigma}_{\alpha}$ being diagonal with the diagonal values equal to $0.4^2$, and $\boldsymbol{\Sigma}_{\xi}$ being diagonal with diagonal values equal to the square of the error budget ($\sigma_i^2 + \tau_i^2$). The covariance matrix of the posterior Gaussian is given by

$$\tilde{\boldsymbol{\Sigma}}_{\alpha} = (\boldsymbol{H}^t \boldsymbol{\Sigma}_{\xi}^{-1} \boldsymbol{H} + \boldsymbol{\Sigma}_{\alpha}^{-1})^{-1}. \tag{7}$$






## 3 Results and Discussion

### 3.1 Validation of the inversion results

The inversion results can be validated by examining how well the posterior flux reproduces the observed mole fractions used in the inversion, which is presented in the Supplement. The median difference between the observed mole fraction and the
prior GEOS-Chem simulation is 1.494 ppb, which is reduced to 0.012 ppb and 0.021 ppb for the hierarchical and analytical posteriors, respectively. In order to further validate the inversion results, a GEOS-Chem simulation with the posterior fluxes from the hierarchical inversion was run. The output from this run was compared to the HIAPER Pole-to-Pole Observations (HIPPO) aircraft data, which was not used to optimise the fluxes in the inversion. This comparison is further discussed in the Supplement, but the median difference between the GEOS-Chem simulation and the observations improves from 1.36 ppb for
the prior to 0.17 ppb for the hierarchical posterior.

### 3.2 Drivers of the surface nitrous oxide growth rate

To investigate the drivers of the observed surface nitrous oxide growth rate (Fig. 4a), we examine the prior (Fig. 4b) and posterior (Fig. 4c) estimates. The only difference between Fig. 4b and Fig. 4c is emissions, demonstrating that emissions are impact the surface growth rate. To investigate the role of the meteorology on growth rate during the last five years, we
ran a forward GEOS-Chem simulation using the prior emissions with repeating 2015 meteorology (Fig. 4d), thus removing any inter-annual meteorological variations. The prior emissions in the prior are constant from 2015 onwards (Table 2), so the only difference between Fig. 4b and Fig. 4d is the meteorology after 2015. Most of the surface growth rate fluctuations after 2016 disappear and the surface growth rate is no longer led by the Southern Hemisphere around 2018, demonstrating that meteorology is a key driver of the surface growth rate. Previous studies have suggested that the Quasi-Biennial Oscillation
(QBO) is responsible (Ray et al., 2020; Ruiz et al., 2021).




**Figure 4.** Atmospheric nitrous oxide surface growth rate for 2011–2020 in four latitude bands and globally, for a. the observations, b. GEOS-Chem with the prior emissions, c. GEOS-Chem with the posterior emissions, and d. GEOS-Chem with repeating 2015 meteorology and prior emissions (which is the same as b. except with constant meteorology after 2015). The growth rates have been smoothed using a LOESS (locally weighted smoothing) algorithm with a span of 0.3.




### 3.3 Nitrous oxide emissions

#### 3.3.1 The global scale

Our posterior mean (and 95% credible interval) of the global mean flux of nitrous oxide for 2011–2020 is 17.2 (16.7–17.7) TgN yr$^{-1}$, with 12.0 (11.2–12.8) TgN yr$^{-1}$ from the land (TransCom regions 0-11 in Fig. S4) and 5.2 (4.5–5.9) TgN yr$^{-1}$ from the
oceans (TransCom regions 12-22 in Fig. S4). These values are within the range of other top-down estimates during this period (Wells et al., 2018; Thompson et al., 2019; Tian et al., 2020; Patra et al., 2022), as shown in Fig. 5a and Fig. 5b. Additionally, the inferred global total emissions show a statistically significant increasing trend over 2011–2020 ($p$-value $< 0.05$ when fitting a classical linear model to the posterior means), as shown in Fig. 5c. This is consistent with previous inversions which have also inferred increasing global emissions (Thompson et al., 2019; Tian et al., 2020; Patra et al., 2022), although this is the first
paper to report emissions for 2020 which are the highest on record. This increase is driven by both land and ocean sectors but we describe further below how partitioning to ocean and land could be influenced by choice of prior.

Imposed on the increasing emissions trend is substantial inter-annual variation, as shown in Fig. 5. Previous studies have found correlation between nitrous oxide fluxes and the El Niño–Southern Oscillation (ENSO) (Ishijima et al., 2009; Thompson et al., 2013; Ji et al., 2019; Patra et al., 2022), with the La Niña phase corresponding to higher nitrous oxide emissions. This
higher emission has been attributed to increased oceanic upwelling bringing up nutrients, which increases primary production, removing oxygen from the subsurface region, which increases denitrification and nitrous oxide production (Stramma et al., 2016; Espinoza-Morriberón et al., 2017; Ji et al., 2019). Soil emissions are also thought to vary with ENSO as a result of changing soil water content and temperature (Ishijima et al., 2009; Saikawa et al., 2013). The ENSO relationship is also seen in our work, where El Niño events in 2014–2016 and 2018–2019 correspond to lower nitrous oxide emissions, although some
of the peaks and troughs in our emissions do occur in different years than in previous studies. For example, previous inversions (Thompson et al., 2019; Patra et al., 2022) infer a peak in emissions in 2013, whereas this work infers a peak in emissions in 2014. These differences are unlikely to be caused by the inversion scheme itself, as performing an analytical inversion rather than using a hierarchical scheme in this work produces the same pattern of inter-annual variability (Fig. 5). The inversions have slightly different prior emissions, but Patra et al. (2022) experimented with using different priors and the inter-annual variability
remained unchanged. It seems the most likely explanation for the disagreement is differences in atmospheric transport between the models. This type of systematic uncertainty is not estimated in any of the inversions presented here.







**Figure 5.** Variation in global annual nitrous oxide emissions over 2011–2020 inferred here and in recent atmospheric inversions (Thompson et al., 2019; Patra et al., 2022), for a. the global land, b. the global ocean, and c. the global total emissions. The shading represents the 95 % credible interval on the mean estimate in this work.



When total emissions are separated into land and ocean contributions, a wide range of emissions are derived by inversions depending on the prior assumptions as shown in Fig. 5a and Fig. 5b. We investigated the sensitivity of the inversion results in the first window (2010–2012) to having the land and ocean priors rescaled to half and double their original values with the results

shown in Fig. 6. Rescaling the prior for either land or ocean results in a redistribution of the nitrous oxide emissions between land and ocean, however the global total is conserved. The redistribution is more marked when rescaling the ocean emissions. This shows that, even in a hierarchical inversion, whilst the global total emissions of nitrous oxide are well constrained by the observations, emissions on a smaller scale are strongly influenced by the prior values, in particular for the ocean regions. However, this range in prior values is not dissimilar to the range used in inversions, for example Patra et al. (2022) uses prior

ocean emissions of 3.4 TgN yr$^{-1}$ whereas one inversion in Thompson et al. (2019) uses a value of over 7 TgN yr$^{-1}$. As a result, different inversion set ups will likely disagree on a regional scale until more observations are available to constrain the fluxes.

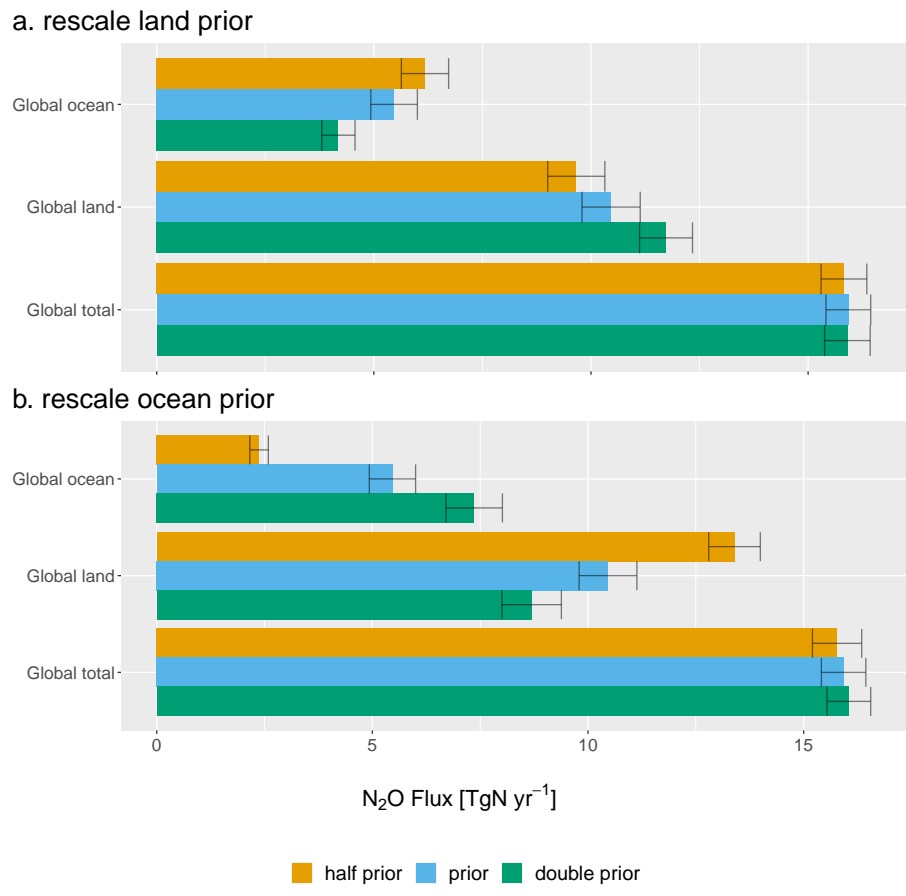

**Figure 6.** The effect of rescaling the prior emissions over a. land and b. ocean on the inferred nitrous oxide flux in the first window of the inversion, inferring fluxes for the year 2011. Orange bars are the inferred fluxes when the prior was halved, blue bars are with the original prior, and green bars when the prior was doubled. The error bars represent the 95 % credible intervals on the estimates.



### 3.3.2 The zonal scale

We focus on zonal inferred emissions, because we believe the problems discussed in Sect. 3.3.1 imply that flux inference on a finer spatial scale is highly challenging with the background network used here, combined with our low-resolution global model setup. Therefore, we do not analyse individual TransCom regions but these are provided in the Supplement with the note that they may be unreliable estimates. The fluxes inferred on a zonal scale are shown in Fig. 7, on both annual and monthly timescales. Moving from the prior to the posterior, there has been a redistribution of emissions, with increased fluxes in the Northern Hemisphere between 0° N and 30° N and reduced fluxes beyond 30° N and 30° S. Most of the increasing global nitrous oxide emissions trend comes from the Northern Hemisphere between 0° N and 30° N, although all zonal bands contribute to the inter-annual variability.

The impact of the hierarchical inversion can be seen by comparing to an analytical inversion within this work, as shown in Fig. 7. In contrast to the well constrained global total, the inversions do infer different zonal totals, with the analytical inversion having a smaller flux and a smaller increasing trend in the Northern Hemisphere between 0° N and 30° N. This difference between the inversions in this zonal band is mainly caused by differences in the Northern African and East Pacific Tropical regions (as shown in the Supplement), which can move further from the prior in the hierarchical inversion (Sect. 3.4).

Whilst it is difficult to directly compare our results to previous inversions which optimise fluxes for different regions and scales, the results are broadly similar. Previous atmospheric inversions also redistribute emissions from the extra-tropics in the prior to the tropics (Thompson et al., 2019; Patra et al., 2022), and assign an increasing trend in emissions to tropical regions, in particular South and East of Asia, Africa, tropical America, and central South America (Thompson et al., 2019; Patra et al., 2022). The main difference in this work is that no trend is derived for Asia and the Americas. This is likely a result of the hierarchical inversion which allows some regions' emissions, in particular North Africa, to be further from the prior if the data dictate it (Sect. 3.4), and hence have a larger emissions trend. In non-hierarchical inversions it appears that the increase in emissions is spread more evenly between regions, perhaps because the prior uncertainty is more homogeneous.

Another notable difference from the prior seen in Fig. 7 is the seasonal cycle in the Northern Hemisphere between 30° N and 90° N, which peaks as winter ends in the posterior, rather than in summer in the prior. This seasonal cycle change has been inferred by other inversions (e.g. Thompson et al., 2014; Nevison et al., 2018; Wells et al., 2018). According to our inversion, the Northern Hemisphere land causes this reversal. The prior anthropogenic emissions only vary on an annual timescale, so the land seasonal cycle predominantly comes from natural soil emissions (Saikawa et al., 2013), which does not account for processes in this latitude band such as freeze-thaw cycles or fertiliser application (Wagner-Riddle et al., 2017; Nevison et al., 2018).




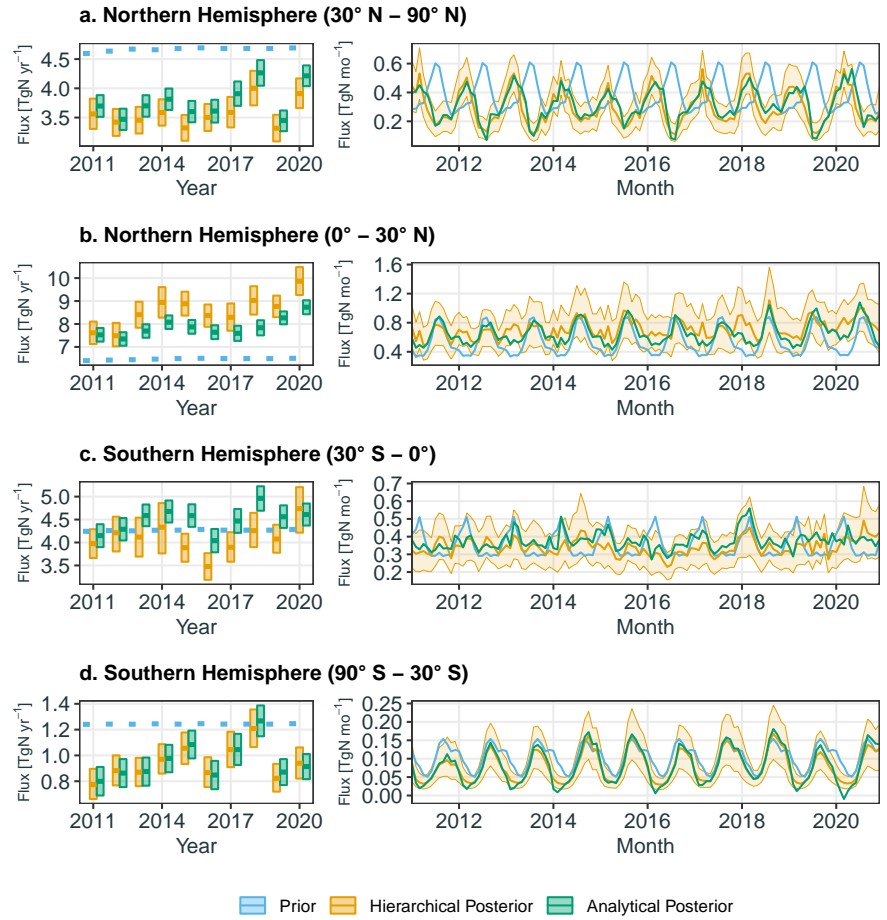

**Figure 7.** Inferred nitrous oxide emissions for 2011–2020 in four zonal bands: a. Northern Hemisphere (30° N – 90° N), b. Northern Hemisphere (0°– 30° N), c. Southern Hemisphere (30° S – 0°), and d. Southern Hemisphere (30° S – 90° S). Left hand side plots show annually averaged emissions and right hand side show monthly emissions, where orange is the hierarchical inversion posterior, green is the analytical inversion posterior, and blue is the prior, with the shading showing the 95 % credible intervals of the hierarchical inversion posterior.

## 3.4 The use of a hierarchical inversion

We used a hierarchical inversion scheme to characterise the uncertainties more objectively compared to previous studies. This was done by including flux scaling factor precisions and error budget scaling factors (Sect. 2.3). The mean values of these hyper-parameters over all windows inferred by the inversion, transformed into standard deviation space, are shown in Fig. 320 8a and Fig. 8b. While only a mean is presented here, the hyper-parameter values are relatively consistent between the years, although this does vary between different regions and stations (see Supplement). The shading in Fig. 8a shows the flux scaling factor standard deviation for each TransCom region. The median value for the flux scaling factor standard deviation is 0.5 (50 %





prior uncertainty), but it is highly dependent on the region. Some regions have a much larger scaling factor standard deviation which means the data provides a strong enough constraint to move these regions far from the prior value. The median value

is very similar to the values commonly imposed through "expert judgment" (0.5–1.0), but the hierarchical inversion scheme infers an uncertainty above 1.0 for every year in two key regions (Eurasia Temperate and Northern Africa). This implies that imposing a strict prior uncertainty of 100 % (or similar) in these regions may overly constrain the prior.

The second type of hyper-parameter shown in Fig. 8b are the error budget scaling factors for each measurement station. This hyper-parameter scales the error budget which includes both a measurement error and model error (Sect. 2.3). Our calculation

of the error budget does not include many other types of error, such as atmospheric transport or chemistry, which the error budget scaling factors can compensate for. The median value of the error budget scaling factor (over all sites and all years) is 1.06, which corresponds to a value of 0.97 in Fig. 8b and a 3% reduction in the error budget, but the values vary substantially by station. This means the error budget in this work is smaller than previous non-hierarchical inversions have imposed. Therefore, on average, previous non-hierarchical inversions are less data-constrained than our framework would suggest. The variation in

the error budget scaling factor between different stations is counter-intuitive, with extra-tropical Southern Hemisphere stations having the largest values, despite small emissions in this area. In this area, the inversion does not match the seasonal cycle and all the inter-annual variation in the observations as well as other areas (shown in the Supplement). The most likely cause is an inadequate prior without enough flexibility to change as a result of solving on the scale of TransCom regions. TransCom regions are particularly restrictive in the Antarctic circle (where the largest error budget scaling factors are), as the TransCom

region for Antarctica also includes Greenland and the Mediterranean Sea (see Fig. S4), limiting the potential for the fluxes in this area to adjust. Another factor could be that the extra-tropical Southern Hemisphere stations generally have lower error budgets before the scaling factor is applied, because of the lower spatial and temporal variability in their mole fractions.

The analytical inversion does not include these hyper-parameters and therefore the uncertainties in the inversion are not as reliable as in the hierarchical inversion. Therefore, the analytical inversion presented in this work should not be interpreted as

an alternative solution, but rather as a way to examine departures from previous work.





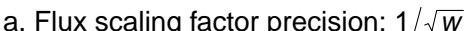

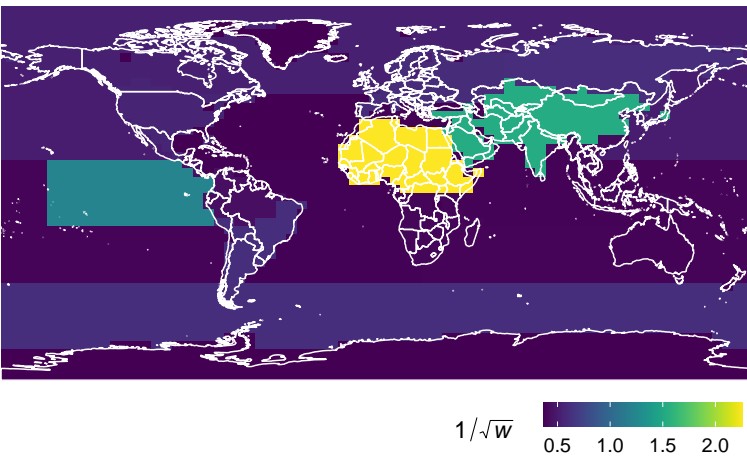

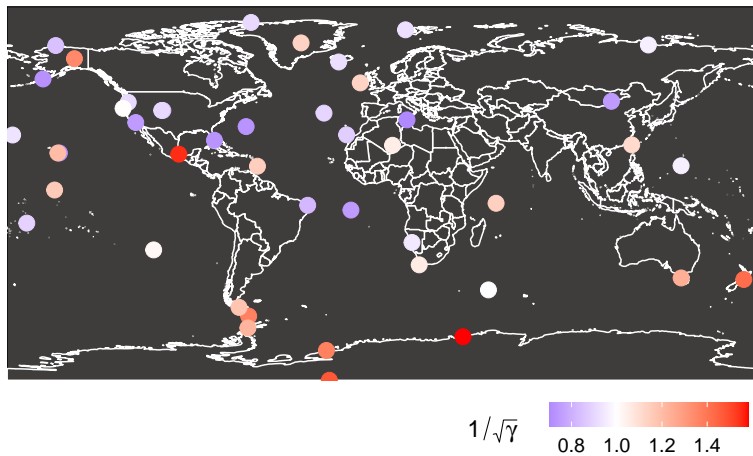

**Figure 8.** The inferred hyper-parameters transformed onto a standard deviation scale. The shading in a. represents the mean uncertainty in the prior emissions ($1/\sqrt{w}$) which is solved for each TransCom region on an annual basis. The coloured dots in b. represent the mean error budget scaling factor ($1/\sqrt{\gamma}$) which is solved for each measurement station.

## 4   Conclusions

We carried out the first hierarchical inversion to solve for global nitrous oxide emissions. We find that global emissions have increased between 2011 and 2020, with substantial inter-annual variability. Emissions derived for 2020 were the highest in this period, 19.5 (95 % credible interval: 18.9–20.1) TgN yr$^{-1}$ due to an increase of emissions in the tropics. On annual timescales, our estimated global emissions differ from other studies, likely due to differences in atmospheric chemical transport models.



We show that the recent atmospheric surface growth rate fluctuations are partly driven by emissions, but also by inter-annual variability in transport.

At the zonal scale, we find several issues with the bottom-up emission estimates used as a prior. The posterior seasonal cycle in the Northern Hemisphere extra-tropics is out of phase with the prior. This may be because the agricultural soil emissions in the prior are only on an annual resolution, and/or because natural soil emissions do not include important processes such as freeze-thaw. Additionally, there has been a substantial redistribution of emissions from the extra-tropics in the prior to the Northern tropics in the posterior. This is the zonal band where most of globally increasing trend is coming from over the time period studied.

By adapting and extending the hierarchical inversion framework of Zammit-Mangion et al. (2022), we have shown that inversions for nitrous oxide can be performed that do not rely on rigid assumptions regarding error budgets or the uncertainty of the fluxes. Our uncertainties are generally smaller than those used in previous studies, and therefore our inversion is more data-constrained. Additionally, our uncertainties vary greatly across different stations and regions, which is not considered in previous non-hierarchical studies. Two innovations in this work over Zammit-Mangion et al. (2022) are the moving window technique, which allow for more efficient computation of fluxes over very long time periods (~10 years or longer), and the incorporation of a truncated Gaussian prior to impose sign constraints on the emissions. The method presented here serves as a framework that can be extended to higher-resolution models (potentially allowing more reliable regional emissions inference) and larger datasets.

*Code and data availability.* The code and data for this work can be found at https://doi.org/10.17605/OSF.IO/SN539. The GEOS-Chem model is available to be downloaded at http://www.geos-chem.org. The atmospheric observations can be obtained from the data providers: https://www.esrl.noaa.gov/gmd/ for NOAA and https://agage.mit.edu for AGAGE.

*Author contributions.* ACS wrote the code (excluding the WOMBAT framework), performed the calculations, and led the writing of the manuscript. MB and AZM wrote the original WOMBAT framework and MB helped develop the framework further for this work. PJF, CMH, PBK, JM, SO, RGP, RFW, and DY provided the AGAGE atmospheric measurements. XL provided the NOAA atmospheric measurements. MM provided ocean model output and helped to interpret the results. ACS, MB, AZM, MR and AG designed the research. All authors contributed to the manuscript.

*Competing interests.* The authors declare that they have no conflict of interest.



*Acknowledgements.* This paper results from research funded by UK Research and Innovation (UKRI) under the South Asian Nitrogen Hub (SANH), grant R100863-101. The project team includes partners from across South Asia and the UK. Neither UKRI nor any of the partner institutions are responsible for the views advanced here. AS was supported by SANH. MB and AZM were supported by the Australian

Research Council (ARC) Discovery Project (DP) DP190100180. The MHD, THD, RPB, SMO, and CGO AGAGE stations are supported by the National Aeronautics and Space Administration (NASA) (grants NNX16AC98G to MIT, and NNX16AC97G and NNX16AC96G to SIO). Support also comes from the UK Department for Business, Energy & Industrial Strategy (BEIS) for MHD, the National Oceanic and Atmospheric Administration (NOAA) for RPB, and the Commonwealth Scientific and Industrial Research Organization (CSIRO) and the Bureau of Meteorology (Australia) for CGO. This work is supported in part by the Cooperative Agreement between NOAA and the

Cooperative Institute for Research in Environmental Sciences (CIRES): NA17OAR4320101.



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
