# Peer review of "Modelling the growth of atmospheric nitrous oxide using a global hierarchical inversion"

_EGUsphere, 2022_

## Author Response (AR2)

**Response to Reviewer 1**

We thank the reviewer for their helpful comments. In this document, we reply to each comment, providing extra detail and outlining how we have updated the manuscript.

**There is no discussion of positive atmospheric growth rate anomaly in 2020 and no mention of the emissions and sink in 2020 and how these may or may not be different with respect to the previous 9 years, although this is alluded to in the abstract (L7). Figure 7 shows the source per latitudinal band for the analytical and hierarchical inversions: the hierarchical inversion indicates a positive source anomaly in 2020 in the band 0-30N, but the source in the analytical inversion is significantly smaller. From the analytical inversion it does not appear that there was any significant source anomaly in 2020. A discussion about how and why the hierarchical and analytical inversion results differ in their source estimates and atmospheric growth rates, especially for 2020, would greatly add to the paper.**

We have added a discussion on the increase in 2020 emissions by adding the following text to line 276 (now line 288):
``Whilst it is difficult to deduce the cause of the emissions increase in 2020 from this study, several factors could play a role. It is likely that natural cycles (e.g. the El Niño–Southern Oscillation (ENSO) (Ishijima et al., 2009; Thompson et al., 2013; Ji et al., 2019; Patra et al., 2022)) contribute to the emissions increase in 2020, alongside the longer term trend in increasing emissions, which has been attributed to a non-linear response of nitrous oxide emissions when nitrogen input is high (Thompson et al., 2019) or an increasing emissions factor due to warming and the redistribution of emissions (Harris et al., 2022)."

A discussion about how and why the hierarchical and analytical inversion results differ is provided in Sections 3.3.2 and 3.4. We have added a discussion of why the 2020 results differ to Section 3.3.1, after line 276 (now line 293):
``The impact of the hierarchical inversion can be seen by comparing to an analytical inversion within this work, as shown in Fig. 5. On the global scale, there is very good agreement in total emissions between the two inversions performed in this work. The only year where the analytical result falls outside of the 95 % credible interval of the hierarchical result is 2020. This is because on a global scale, nitrous oxide emissions are well constrained by the observations so the inversions give consistent solutions. However, there are fewer observations to constrain the emissions in 2020 (as 2021 observations were not available). The hierarchical inversion moves further from the prior because the uncertainties in the inversion can be adjusted (Sect. 3.4). The two inversions do not agree on the land and ocean emissions for 2011--2020 as well as for the global total emissions over the same time period. The land and ocean emissions are less constrained by the observations than the global total emissions, and so differences in the uncertainties in the inversions (Sect. 3.4) lead to different results. This is also the case for the zonal emissions which are discussed in Sect. 3.3.2."

**L4: The authors should replace "non-Gaussian" with "truncated Gaussian", which is what is described in section 2.3 for the prior uncertainties, and "Gaussian" for the model-measurement discrepancies as described in section 2.4.**

Thank you for pointing this out, it has been changed as suggested.

**L24: The reference Solazzo et al. 2021 is not really appropriate here since it is the natural sources of N2O that are being discussed, whereas Solazzo et al. discuss only anthropogenic emission estimates derived using emission factor approaches.**

We agree and the reference to Solazzo et al. has been removed.

**L29: It sounds as though NOAA and AGAGE are the only ones measuring N2O, which is not the case. Please at least add "among other laboratories" (e.g. ICOS network, which has continuous measurements of N2O since about 2018).**

We agree and have rephrased this sentence to read:
``The atmospheric abundance of nitrous oxide is monitored by several laboratories, and in this work we use measurements taken by the National Oceanic and Atmospheric Administration (NOAA) (Dlugokencky et al., 2021; Sweeney et al., 2021) and the Advanced Global Atmospheric Gases Experiment (AGAGE) (Prinn et al., 2000, 2018)."

**L33: It is not "meteorology" that is driving the growth rate, rather (and this is the conclusion of Ruiz et al.) climate oscillations, in particular, the Quasi-Biennial Oscillation, are an important (but not the only) driver of variability in the tropospheric growth rate of N2O.**

``Meteorology" has been replaced by ``climatic variability".

**L67: N2O fluxes on land can be negative, however, the sink is thought to be rather small (Tian et al. 2020). Therefore, please change the end of this sentence to "we expect land emissions to be predominantly positive"**

We have rephrased this to ``we expect land emissions to be predominantly positive", as the land sink is very small.

**L143: According to Eq. 1, excluding the error term, the true flux is modelled as the sum of the prior flux and a scaling of the basis function (phi) (not a scaling of the prior flux)**

The basis functions are the prior fluxes in the different TransCom regions, as stated on line 142 (now line 146), so the original phrasing is correct.

**L260: There is no "record" of the emissions, but only of the growth rate. Please change this to "the first paper to report emissions for 2020, the year which had a record growth rate"**

This has been rephrased to:
``... this is the first paper to report emissions for 2020 which are likely to be the highest in 2011--2020."

**L310: In this study, the posterior fluxes in the NH, 30-90N peak in January-February, which is earlier than found in the cited previous studies, which find a maximum in spring, around March-April. I strongly suspect that this winter peak in emissions is due to model-transport errors, and Fig. S1 shows that the model does not capture the phase of the seasonal cycle in atmospheric N2O in the northern latitudes (phase is approximately 6 months out of phase with the observations), although some of this mismatch may be due to the missing or incorrect seasonality in the prior fluxes. A winter (January-February) maximum in the emissions for the latitudes 30-90N is very difficult to reconcile with what is understood about the drivers of the emissions, which include management (e.g., timing of fertilizer application) and environmental factors (e.g., soil moisture and temperature).**

In this study, the posterior fluxes in the NH 30-90N typically peak in March, in agreement with the cited previous studies. This has now been explicitly stated in line 310 (now line 336). Additional gridlines have also been placed on the figure to make this easier to read.

**L333: The authors say that their error budget for observation-space uncertainties is smaller than in previous variational inversions, but the observation number and frequency between this study (monthly observations) and other inversions (hourly or afternoon averages) is very different and thus the observation-space uncertainties also need to be different to reflect this.**

We agree and this has been rephrased to:
``This means the error budget in this work is smaller than a non-hierarchical inversion would have imposed. Therefore, a non-hierarchical inversion for the same number of data points and uncertain parameters would be less data-constrained than our framework."

**L335: I think the result for the error budget scalar in the extra-tropical SH is not that surprising considering the large model-observation differences there.**

We agree, but starting with no prior knowledge of the system, we would describe it as ``counter-intuitive". This has been clarified by rephrasing to ``somewhat counter-intuitive".

**L337: I think the authors mean "or the inter-annual variation" not "all the inter-annual variation"**

This has been changed as suggested.

**L338: Concerning the reason for the cause of poorer agreement with the observed seasonal cycle and interannual variability for the extra-tropical SH, this is very likely also due to the large error budget scaling factors in this region, which means that the observational constraint is weaker. Furthermore, the Antarctic region (i.e., Transcom region T00) has very likely negligible emissions, and the variability in atmospheric N2O at**

the extra-tropical SH sites is driven by atmospheric transport, including stratosphere-troposphere exchange, ocean fluxes, and to a smaller extent fluxes over the small amount of land in the SH extra-tropics.

The error budget scaling factors are estimated in the inversion, so the reverse explanation for the poor agreement in the extra-tropical SH is also possible: that the error budget scaling factors are larger because of the poorer agreement. We have rephrased and added to this section to include the importance of atmospheric transport:
``One of the most likely causes of the large error budget scaling factors and observational mismatch is an inadequate prior without enough flexibility to change as a result of solving on the scale of TransCom regions. TransCom regions are particularly restrictive in the Antarctic circle (where the largest error budget scaling factors are found), as the TransCom region for Antarctica also includes Greenland and the Mediterranean Sea (see Fig. S1), limiting the potential for the fluxes in this area to adjust. Another factor causing the large error budget scaling factors and observational mismatch could be that the extra-tropical Southern Hemisphere stations generally have lower error budgets before the scaling factor is applied, because of the lower spatial and temporal variability in their mole fractions. Additionally, because of the low emissions in this area, the variations in atmospheric nitrous oxide mole fractions are mainly driven by atmospheric transport, which the inversion cannot adjust."

**L351 (and L5-6): The statement "we show that the recent atmospheric surface growth rate fluctuations are partly driven by emissions but also by inter-annual variability in transport" is not very well supported. The authors do not discuss or quantify the contribution of variability in emissions to that in the atmospheric growth rate, nor do they quantify the contribution from inter-annual variability in transport.**

This is demonstrated in Fig. 4 and discussed in L242-250 (now lines 250-259). Running the model with repeating 2015 meteorology changes the atmospheric growth rate, as does changing the emissions. Indeed, we have not quantified the exact contributions, but we have amended the text slightly to ``we show that the recent atmospheric surface growth rate fluctuations are likely to be driven both by emissions and also by inter-annual variability in transport".

**L354: Concerning the phase of the posterior seasonal cycle in the NH, I think this result may not be very robust given the uncertainties in the modelled atmospheric transport, therefore, the authors should include as a reason for this shift in phase "errors in modelled atmospheric transport".**

We believe this comment comes from the reviewer's misreading of the seasonal maximum in their comment about L310, and as such is an unnecessary change. We have clarified the misunderstanding in the response to the comment about L310.

**L361: The statement that this inversion is "more data constrained" compared to previous studies is not quite true. The observation constraint is not only determined by the observation uncertainties but also the number of observations, and there are previous examples of N2O inversions using vastly greater numbers of observations (e.g. afternoon mean observations).**

We agree and have rephrased this to:
``Our uncertainties are estimated by the inversion are generally smaller than those that would be used in a non-hierarchical inversion for the same number of data points and uncertain parameters, and therefore our inversion is more data-constrained."

**Response to Reviewer 2**

We would like to thank the reviewer for their helpful comments. In this document, we reply to each comment, providing extra detail and outlining how we have updated the manuscript.

**Line 27-29: There are other networks that have routine measurements of N2O. Why not mention or include them in this work?**

We agree and have rephrased this sentence to read:
``The atmospheric abundance of nitrous oxide is monitored by several laboratories, and in this work we use measurements taken by the National Oceanic and Atmospheric Administration (NOAA) (Dlugokencky et al., 2021; Sweeney et al., 2021) and the Advanced Global Atmospheric Gases Experiment (AGAGE) (Prinn et al., 2000, 2018)."

**Line 44-45: This statement makes it seem like additional observations are the main requirement for constraining regional inversions, but that is not problem being addressed by this work. Consider rephrasing here to better motivate what's actually being done in this study.**

This has been rephrased to:
``These discrepancies suggest that new measurement or modelling approaches are required to constrain fluxes at the regional scale in global inversions."

**Line 58-67: This paragraph is helpful in that it mentions some of the shortcomings of past work that might be addressed in this study, but I think it could use a more definitive take-home statement about what is being done here and how the main findings will be presented. This way the reader will have a clearer idea of what to expect before they read on.**

The following has been added to this paragraph:
``In this work, we investigate nitrous oxide emissions on a global and zonal scale using the hierarchical inversion. To help examine departures from previous inversions and explore the benefits of the hierarchical framework, we compare to results from an analytical inversion."

**Line 67: "as we expect land emissions to be only positive." Ok, I guess by definition emissions are positive, but I think soils can also be a sink for N2O?**

We have rephrased this to "we expect land emissions to be predominantly positive", as the land sink is very small on the scales being investigated.

**Section 2.1: There are other networks that measure N2O, did none of those data meet the criteria to be included in this study?**

There are potentially a small number of other sites in other networks that could have made our selection criteria. However, many of these stations are located in the same or similar places as the NOAA/AGAGE networks so do not add much independent information to the inversion. Whilst developing the current inversion, we did experiment with including/excluding slightly different sites and there was not a substantial change in the emissions inferred. This is likely because the main focus of our paper is global and zonal scales, rather than regional scales, where additional measurements will be important.

**Line 109: "starting from an atmosphere with a constant nitrous oxide model fraction." Does this constant mole fraction extend into the stratosphere as well? If so, has any analysis been done to see how well the model represents stratospheric N2O? It not, do the authors expect their spin-up approach to affect the results, given that stratosphere-troposphere exchange drives the seasonality of surface N2O at remote sites away from continental sources?**

This constant mole fraction does extend into the stratosphere, but we have checked that stratospheric nitrous oxide is adequately represented. This was done by examining the simulated zonal mean atmospheric nitrous oxide distribution with height as well as the lifetime. We have expanded the sentence starting on line 109 (now line 112) to make this clearer:
``The resulting initial condition field matches surface nitrous oxide observations to within a few ppb, has a zonal and annual mean latitude–altitude cross section of nitrous oxide mixing ratio that matches other models (Thompson et al. (2014b), and also gives a nitrous oxide lifetime of 120 years, in good agreement with Ko et al. (2013) and Prather et al. (2015)."

**Line 131: In the abstract and introduction the authors describe the prior emissions as non-Gaussian, but here they are described as Gaussian truncated at zero. Please update the earlier descriptions to be consistent.**

This has been done as suggested.

**Line 132: Here the authors mention that they use a 3-year moving window to reduce computational cost of the inversion. It would be really helpful if the authors included more information about the computational cost of this approach versus analytical or 4D-Var, for example, to put this statement in context.**

We have added some information about the computation time to line 204-5 (now lines 210-212):
``The method of Zammit-Mangion et al. (2022) is too computationally expensive to run over the long time period in this study. We instead use a moving window approach, which reduces the computation time from weeks to days, but is still much longer than the seconds it would take to solve analytically."

**Line 142-143: "the true flux in a region is modelled as a scaling of the prior flux in that region." But, equation (1) shows the true flux as a sum of the prior flux and a scaling of the basis function, so I'm unclear what exactly is begin done. If equation (1) is correct, it seems the prior scaling factor would be zero, and then positive or negative posterior scaling factors are allowed? Is that correct? Note: I found the answer to this question later in Section 2.3.4, where it states that the scaling factors are greater or equal to -1. Suggest moving this info earlier, or at least pointing the reader to Section 2.3.4 here.**

This has been added as suggested.

**Line 153-156: Here in the description of the response function construction it would be a helpful to talk more about the computational cost of the inversion. It sounds like it requires 3,036 perturbed forward runs of the model? How long does that take? How does it compare to the cost of the analytical inversion?**

Yes, 3,036 perturbations to the model are required, as well as an original ``base" run. The perturbed simulations are used to construct the basis functions used in both the analytical and the hierarchical inversions.

Apart from the ``base" simulation, which must be run first to derive the initial conditions, the perturbed simulations can all be run in parallel, greatly reducing the computation time. The computational burden can be further reduced by using tagged tracers within the GEOS-Chem model for emissions from each of the TransCom regions. With these tweaks, only 132 perturbed forward simulations are required. On our HPC system, the ``base" simulation took about a week, followed by another week for all the perturbed simulations.

To express this within the paper we have added the following:
``Running the perturbed simulations is computationally expensive, but can be reduced by running simulations in parallel and using tagged tracers within GEOS-Chem for the emissions from the different TransCom regions. These model runs are required for both an analytical inversion as well as the hierarchical inversion."

**Line 163: Can the authors clarify what is meant here by "similar error properties"?**

This phrase has been removed as it is unnecessary.

**Figure 3: Could some text be added to the graphical model to clarify the processes that are represented by the different arrows? I don't find the model particularly helpful to the reader as is.**

The arrows, as is customary for graphical models, do not represent processes but statistical dependence between the nodes (random variables). This explanation has now been added to the caption:
``The arrows represent the statistical dependence between the variables."

**Lines 204-209: Here it is mentioned that the moving window was used for computational efficiency for long-term inversions. Can the authors quantify how much computational cost is saved using this approach?**

This has been done as specified in the response to the comment about line 132.

**Lines 242-250: It sounds like the authors are concluding that interannual variability in the N2O growth rate is driven by meteorology from their test in Fig. 4d. The authors mention the QBO as a potential driver, and I think this discussion could be expanded upon (by citing additional peer-reviewed literature) to elucidate the processes that would impact surface N2O year-to-year. For example, do different phases of the QBO result in different rates of strat-trop exchange, or does it more have to do with differences in horizontal transport in the atmosphere?**

We have expanded this to read:
``Previous studies have suggested that the Quasi-Biennial Oscillation (QBO) is an important driver of the nitrous oxide growth rate, as it modulates the stratosphere to troposphere mass flux (Ray et al., 2020; Ruiz et al., 2021)."

**Line 260: The authors should avoid overinterpreting the 2020 results here, given that they are less constrained than other years in the inversion.**

We have mentioned that this year is less constrained in line 209 (now line 216), but this should be accounted for in the uncertainty in the emissions, and the emissions are still well constrained on a global scale. When considering the emissions split between land and ocean, the larger uncertainty for 2020 can be seen in Fig. 5. We have rephrased the line in the paper to:
``... although this is the first paper to report emissions for 2020 which are likely to be the highest in 2011--2020."

**Figure 7: In the monthly plots it's quite difficult to discern the timing of the peaks. Suggest making these plots larger or at the very least adding more tick marks to denote the years and months.**

We have added annual gridlines and it is now stated which month the seasonal maximum occurs in the text for Northern extratropics.

**Line 337: "The most likely cause is an inadequate prior…" Other studies have attributed the poor model representation (and, thus, high error scaling factors in this work) of southern hemisphere N2O to uncertainties in model transport. Can the authors discuss this possibility here?**

We have altered this as suggested:
``One of the most likely causes of the large error budget scaling factors and observational mismatch is an inadequate prior without enough flexibility to change as a result of solving on the scale of TransCom regions. TransCom regions are particularly restrictive in the Antarctic circle (where the largest error budget scaling factors are found), as the TransCom region for Antarctica also includes Greenland and the Mediterranean Sea (see Fig. S1), limiting the potential for the fluxes in this area to adjust. Another factor could be that the extra-tropical Southern Hemisphere stations generally have lower error budgets before the scaling factor is applied, because of the lower spatial and temporal variability in their mole fractions. Additionally, because of the low emissions in this area, the variations in atmospheric nitrous oxide mole fractions are mainly driven by atmospheric transport, which the inversion cannot adjust."

**Line 350: "Likely due to different chemical transport models…" I think the authors could do more to discuss the differences between their work and other studies. If they are just due to differences in chemical transport models (which I assume means differences in their representation of model transport?) then what is this different inversion framework adding? Might we expect the results to be different from other studies using the hierarchical inversion?**

Our hierarchical and analytical inversions use the same transport model with somewhat different inversion methods and get similar results. This suggests that if a transport model from another study was used, we might get similar global and zonal results to that study. However, there will also be differences caused by using different optimisation regions and prior means. There are greater differences between our two inversions on a regional scale (as seen in the Supplement), but we chose not to discuss specific regions due to the problems described in Sect. 3.3.1. What the hierarchical inversion adds to the study is described in detail in Sect. 3.4 (uncertainties that are informed by the data and vary greatly across different stations and regions). Whilst, for global scale inversions of nitrous oxide emissions, the hierarchical and analytical inversion produce similar global and zonal emissions, that is not necessarily true for other gases and scales of inversion.

This has been rephrased to:
"... likely due to differences in atmospheric chemical transport models and optimising emissions for different regions, rather than the inversion method."

**Lines 353-358: The findings listed in this paragraph are all broadly consistent with what other inversion studies have found. Suggest instead highlighting the new findings of this work and what recommendations can be made from the error budgets derived (e.g., which regions are the least constrained by our global N2O observing network?)**

Whilst our conclusions are indeed broadly similar to other studies, we believe they are still worth stating here. It is hard to say which regions are the least constrained, because the likely answer is the regions which have stayed very close to the prior, but alternatively, the prior could just been correct in those regions. We chose not to discuss specific regions due to the problems described in Sect. 3.3.1.

**Lines 361-362: "therefore our inversion is more data-constrained." I'm not sure the authors have demonstrated this here. For one, they use fewer observations than other N2O inversion studies have used. Also, does a lower prior observational uncertainty budget necessarily mean a higher data constraint? I typically think of data constraints as being quantified via posterior error reduction or degrees of freedom from the inversion. Can these kind of metrics be calculated from the hierarchical inversion framework?**

We agree and have rephrased this to:
``Our uncertainties are estimated by the inversion are generally smaller than those that would be used in a non-hierarchical inversion for the same number of data points and uncertain parameters, and therefore our inversion is more data-constrained."

**Line 19-20: This sentence is awkward. Consider revising.**

This has been rephrased to:
``Additionally, nitrous oxide is currently the largest contributor to stratospheric ozone depletion, when considering ozone depletion potential-weighted anthropogenic emissions (Ravishankara et al., 2009)."

**Line 137: Consider making Fig. S4 into Fig. S1 since it is the first supplemental figure referenced.**

This has been done as suggested.

**Line 190: Delete second instance of "uncertainty" here.**

This has been done as suggested.

**Line 244: Change "are impact" to "are impacting"**

This has been done as suggested.

**Response to Editor**

**Thank you for submitting a revised manuscript. The response to the reviewers' comments and revisions to the manuscript is largely satisfactory. I accept the revised version for publication, subject to one small change. Both reviewers asked about other laboratories performing N2O measurements and why they were not included in your work. In your response to Referee #2, you responded that you initially tried adding data from sites other than NOAA and AGAGE and did not see any substantial change in your results. I think this is important information to include in the paper. I suggest that you add a brief explanation of what other data was tested and why you decided not to use it.**

We only tried including other sites from the NOAA network, not from other networks. We have now included this within the paper, on what is now line 81:

"Whilst this filtering was somewhat subjective, our results were not substantially changed if we included additional sites that appeared to experience moderate regional influence."